# Design and Characterization of Semi-Floating-Gate Synaptic Transistor

**DOI:** 10.3390/mi10010032

**Published:** 2019-01-07

**Authors:** Yongbeom Cho, Jae Yoon Lee, Eunseon Yu, Jae-Hee Han, Myung-Hyun Baek, Seongjae Cho, Byung-Gook Park

**Affiliations:** 1Department of Electronics Engineering, Gachon University, Gyeonggi-do 13120, Korea; jj2928@naver.com (Y.C.); ldhh1015@nate.com (J.Y.L.); yesemic@naver.com (E.Y.); 2Department of Energy IT, Gachon University, Gyeonggi-do 13120, Korea; jhhan388@gachon.ac.kr; 3Department of Electrical and Computer Engineering, Seoul National University, Seoul 08826, Korea; applewhisky90@gmail.com

**Keywords:** semi-floating gate, synaptic transistor, neuromorphic system, spike-timing-dependent plasticity (STDP), highly miniaturized transistor structure, low power consumption

## Abstract

In this work, a study on a semi-floating-gate synaptic transistor (SFGST) is performed to verify its feasibility in the more energy-efficient hardware-driven neuromorphic system. To realize short- and long-term potentiation (STP/LTP) in the SFGST, a poly-Si semi-floating gate (SFG) and a SiN charge-trap layer are utilized, respectively. When an adequate number of holes are accumulated in the SFG, they are injected into the nitride charge-trap layer by the Fowler–Nordheim tunneling mechanism. Moreover, since the SFG is charged by an embedded tunneling field-effect transistor existing between the channel and the drain junction when the post-synaptic spike occurs after the pre-synaptic spike, and vice versa, the SFG is discharged by the diode when the post-synaptic spike takes place before the pre-synaptic spike. This indicates that the SFGST can attain STP/LTP and spike-timing-dependent plasticity behaviors. These characteristics of the SFGST in the highly miniaturized transistor structure can contribute to the neuromorphic chip such that the total system may operate as fast as the human brain with low power consumption and high integration density.

## 1. Introduction 

In 2016, AlphaGo, one of the results of artificial intelligence (AI) won the Go game against top-ranked Go players [1]. Because Go had been considered suitable only for humans, as it requires not only intelligence but also experience, this achievement led to a media sensation. Why has AI now attracted the public’s attention, and why has AI research become so active once again? One of the reasons is the efficiency of the AI system. Currently, as AI technology develops, a method for increasing operation speed by using a graphic card in parallel is being adopted [2]. Hence, although the amount of necessary computation is large, replicating and mimicking activities that humans would carry out for human mental activities have become possible. So far, these operations have been realized in the software technologies in the von Neumann architecture. However, in order to imitate the human brain with higher resemblance, which performs great deal of mental activities with very small amount of power consumption, power efficiency should be considered more importantly now and in the future calling for the hardware-driven neuromorphic system. In other words, efficiency should be supported not only by algorithms but also by the hardware. In this respect, studies on neuromorphic chips that integrate software and hardware are attracting particular interest [3,4,5,6]. 

In this study, we have focused on synaptic cells shown in Figure 1, which is thought to be closely related to experiences in human mental activity and accumulation of them, as the element to imitate the nervous system [7,8,9,10]. This biological motivation is projected to an electronic component, synaptic transistor. In order to enable low-power and high-speed operations, a poly-Si semi-floating gate (SFG) structure with a tunneling field-effect transistor is adopted for realizing short-term potentiation (STP), and a SiN charge-trap layer is stacked on the SFG for realizing long-term potentiation (LTP) operation. Further, we have obtained the spike-timing-dependent plasticity (STDP) characteristic [11,12,13]. Finally, we propose a novel synaptic device that has STP/LTP capabilities with STDP operation which are the essential functions of the human biological synapse. 

## 2. Device Structure and Operation Schemes 

Figure 2a shows the schematic of the proposed SFG synaptic transistor (SFGST) and its circuit symbol representation. Although the proposed device is based on the integration of volatile and nonvolatile memory components in a miniaturized transistor, write/erase operations are analogously termed as potentiation and depression from the stance of new synaptic functions expected from the proposed device. As shown in Figure 2b, the n^+^/n/p^+^ (drain/channel/SFG) junction is embedded for low-power and high-speed potentiation by hole tunneling into the SFG. When the first and second gates are negatively biased and the drain is positively biased, potentiation occurs as demonstrated in Figure 2c. The device fabrication has full Si processing compatibility and higher mass producibility. Hole tunneling takes place between the drain junction of the SFGST and the first gate by the operation of a p-type tunnel field-effect transistor (TFET), by which holes are accumulated in the SFG. This accumulation of holes in the SFG lowers the threshold voltage of the SFGST and increases the channel conductivity eventually. These series of changes in carrier population and potential distribution make up the potentiation process. When the biases on the two terminals are reversed, the holes are discharged from the SFG by drift and diffusion due to the turn-on of the diode part residing between the drain junction of the SFGST and the SFG. Equivalently, it can be understood as electrons are charged in the SFG and the threshold voltage of the SFGST is elevated. These series of carrier and potential redistributions make the depression process happen. Figure 3 and Table 1 show the mesh structure and the device parameters of the simulated device by technology computer-aided design (TCAD) [14]. The meshes are weaved more densely in the SFG, nitride charge-trap layer, and near the tunneling sites for achieving higher accuracy in this simulation. In order to obtain the TCAD simulation results with higher accuracy and credibility, multiple physical models including Fowler–Nordheim (FN) tunneling model, band-to-band tunneling model, nitride charge-trap model, concentration-dependent generation-recombination model, and concentration/temperature-dependent mobility models have been activated simultaneously for respective simulation tasks. The band-to-band tunneling model has been adjusted with the correction factors empirically suggested by Hurks [15]. 

## 3. Synaptic Operation Characteristics 

### 3.1. Short-Term and Long-Term Potentiation Operations

SFG is partially connected to the channel at the end unlike the commonly used floating gates which is isolated from the channel. By using the SFG, holes can be easily stored by the tunneling current and erased by the drift and diffusion mechanisms. Here, the holes accumulated in the SFG region by proper operation voltages but vanish if there is no hold bias. This characteristic can be adopted for realizing the STP operation. However, when input pulses are successively provided before the holes vanish, the total charges in the SFG increase with time. The number of holes in the SFG increases as the pulses with short time interval are successively applied and the number of newly generated holes is larger than that of holes disappearing by either diffusion or recombination. The higher energy states allowed in the SFG region were mostly vacant due to smaller occupation probabilities since they are located in the tail region of the Fermi–Dirac distribution but now they are occupied by the holes accumulated in a large number in the SFG. The holes in the higher energy states have higher probability of Fowler–Nordheim (FN) tunneling into the nitride charge-trap layer through the tunneling oxide energy barrier deformed to a triangular shape under a high electric field. Thus, the FN tunneling has the predominance in the region of large amount of holes accumulated in the SFG as shown in Figure 4a. These characteristics make distinction between STP and LTP. Figure 4b shows the actual results of the simulated total charges in the SFG (left) and nitride (right) regions after successive potentiating pulses. Here, the charges in the specified regions identify the total net charges which have been extracted by integration of current over a period of time for an operation. It should be reasonable to have an individual look into the electron and hole densities in order to investigate the time-varying amount of stored charges in case of conventional floating-gate (FG) memory devices. The proposed device in this work equips an SFG but there should be conduction of electrons and hole into and out of the floating gate according to the relation among potential distributions over the diode and TFET regions linked to the floating gate. Thus, total net charge might make a more practical sense in this case and Figure 4b conveys the total net charges vs. number of potentiation pulses. Here, a negative value implies that the electrons have the predominance in population, and inversely, a positive one reveals the predominance of holes. It is confirmed that more than three pulses are required for the transition from STP to LTP at the bias condition of *V*_GS1_ = *V*_GS2_ = −1.5 V with a pulse width and interval of 1 μs. It is expected that an increased number of pulses will be required for STP to transit into LTP as the tunneling oxide (TO) becomes thicker. The hole current density after a specific number of pulses is shown in Figure 5. Here, it is Figure 5 that qualitatively demonstrates the directions of carrier movements over the short- and long-term potentiation processes. Holes are injected by the operation of p-type TFET functional region near the drain for potentiation. The holes are injected into the SFG and a part of them occupying higher energy states in the Fermi–Dirac distribution after accumulation of significant amount of holes tunnel into the nitride layer.

Figure 6a shows the hole distributions in the simulated synaptic transistor after 1 and 20 potentiation pulses are applied. The electric field across the blocking oxide, nitride charge-trap layer, tunneling oxide, and SFG along the cutline A-A′ is investigated in Figure 6b. It is confirmed by Figure 6a,b that the charge-trap layer between tunneling oxide and blocking oxide layers has relatively larger population of holes injected from the SFG by tunneling. Consequently, the electric field across the cutline is increased, with the reference at point A′, owing to the holes trapped in the nitride layer. 

Figure 7 shows the retention characteristics under a constant read bias condition, *V*_GS1_ = *V*_GS2_ = *V*_DS_ = 0.5 V, after different numbers of pulses are provided. When the number of pulses is 0, 1, and 3, the drain current decreases as time passes, and then, converges into the initial state due to the semi-floating structural characteristic. In contrast, when the number of pulses is more than 3, such as 10, 20, and 50, the drain current converges into a higher value than that of the initial state. In particular, when the number of pulses exceeds 50, the current levels of STP and LTP are almost the same. As previously mentioned, this is due to the trapped charges in the nitride layer and the electric field repulsing the holes downward. Further, in both the short-term and long-term potentiation cases, multi-level states can be realized. Hence, it is confirmed that SFGST can distinguish between STP and LTP with multi-level current states, which is the essence for mimicking synaptic operation which modulates the connectivity strength by frequencies. While performing the STP operations, the accumulated holes vanish by recombination and diffusion conduction. When the holes are still existing in the SFG, the threshold voltage of the SFGST is elevated and the sensing current increases in accordance. On the other hand, the number of holes in the SFG decreases as time passes without successive pulses with a short enough interval time and the sensing current goes back to the original low level as the result. Thus, the pulse time shorter than times for recombination and travelling by diffusion can only fluctuate the SFG potential and the sensing current. However, in the existence of a large amount of holes accumulated in the SFG, without being provided with a long enough pulsing time to be released from the SFG, the holes become very probable to occupy higher energy states, see a lower effective energy barrier toward the nitride charge-trap layer, and can subsequently tunnel into nitride even at a smaller tunneling electric field necessitating a small voltage. Once the holes are trapped in the nitride, the sensing current is semi-permanently decided and invariant with time as shown in Figure 7. The additional pulses under the LTP condition contribute to increasing the amount of holes trapped in the nitride and determine the level of constant current, which eventually modulates the electrical conductivity of the SFGST and presents the multi-level states. 

### 3.2. Spike-Timing-Dependent Plasticity (STDP) 

In order to utilize the SFGST as a synaptic device capable of STDP operation, the array architecture for realizing the artificial spike neural network (SNN) hardware based on the proposed SFGST device with the full accommodation of the operation bias schemes needs to be proposed, as demonstrated in Figure 8. Since the second gate and the drain are tied together, the device is operated as a three-terminal device. Here, the operating condition that the biases of the first and the second gates are opposite to each other makes it possible to operate the SFGST realizing the STDP behavior. Figure 9 shows the simulated transient STDP characteristics of the SFGST for a single triangular spike. If the pre-neuron signal comes in earlier than the post-neuron signal, the time difference has a positive value, and vice versa. It is confirmed that the SFGST follows the Hebbian learning rule successfully. The change in weight increases as |Δ*t*| decreases, and decreases as |Δ*t*| increases. 

This can be also verified by Figure 10, which shows the variation in the threshold voltage by the potentiated or depressed SFG. As briefly mentioned earlier, the variation in the threshold voltage becomes larger as |Δ*t*| gets larger. Finally, Figure 11 shows the simulated learning operations as a function of number of potentiation pulses. From the results demonstrating that the given pre- and post-neuron signals potentiate the SFGST making the distinction between STP and LTP, it is confirmed that the designed SFGST is fully functional as a synaptic device. 

## 4. Conclusions

In this work, a novel synaptic transistor featuring the semi-floating gate and charge-trap layer has been proposed and designed, and its essential synaptic operations have been verified through TCAD simulation. The SFGST performs both STP and LTP operations discriminable by the number of potentiation pulses. Also, it is confirmed that multiple states, i.e., multiple conductance values can be obtained in the LTP, which corresponds to modulation in the biological synaptic connectivity representing the synaptic weight. Based on the STP and LTP operation capabilities, STDP operation has been verified and the presumable array architecture into which the proposed synaptic transistor and the operation schemes are converged has been proposed. The proposed miniaturized transistor embedding both volatile and nonvolatile memory components can be a promising intelligent component realizing the hardware-driven neuromorphic system which is mainly based on the semiconductor technology with full Si processing compatibility. 

## 5. Patents

(1)Seongjae Cho and Yongbeom Cho, “Synaptic Semiconductor Device and Neural Networks Using the Same,”
-Korean patent filed, 10-2017-0152803, 16 November 2017-United States patent filed, 15/892,658, February 2018.(2)Byung–Gook Park and Seongjae Cho, “Neuron circuit and synapse array integrated circuit architecture and fabrication method of the same,”
-Korean patent filed, 10-2017-0062097, 19 May 2017.-United States patent filed, 15/895,255, 13 February 2018.

## Figures and Tables

**Figure 1 micromachines-10-00032-f001:**
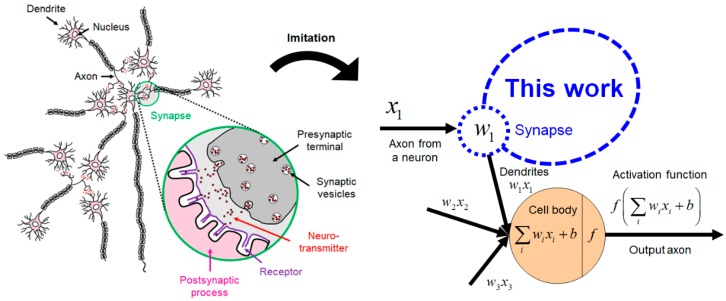
Biological nerve cell element targeted to imitate by the electron device in this work and its mathematical representation.

**Figure 2 micromachines-10-00032-f002:**
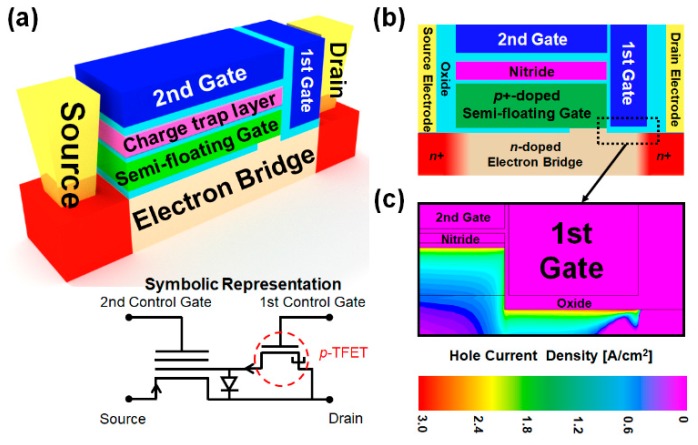
Device structure and potentiation process. (**a**) Aerial view of the proposed synaptic device and its circuit symbol representation; (**b**) Cross-sectional view of the device; (**c**) Contour of hole current density during the potentiation through band-to-band tunneling.

**Figure 3 micromachines-10-00032-f003:**
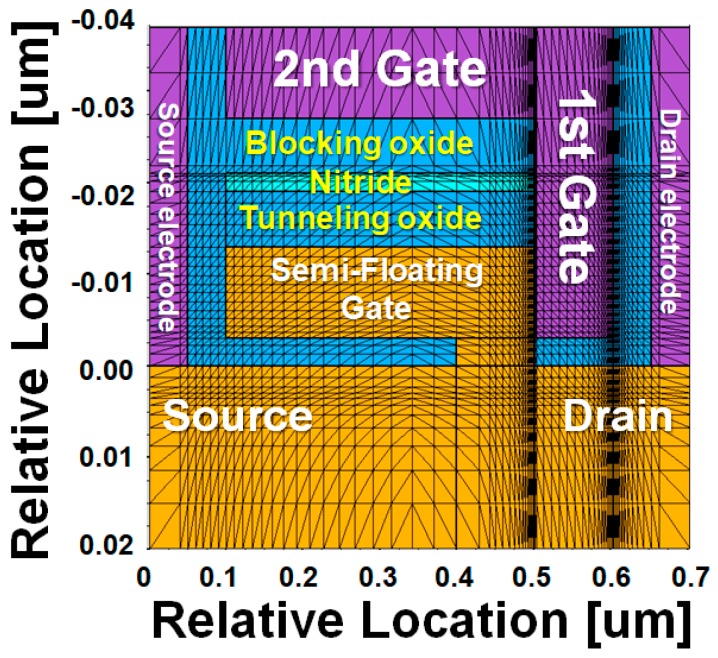
Mesh structure of the simulated device with notations of the terminals.

**Figure 4 micromachines-10-00032-f004:**
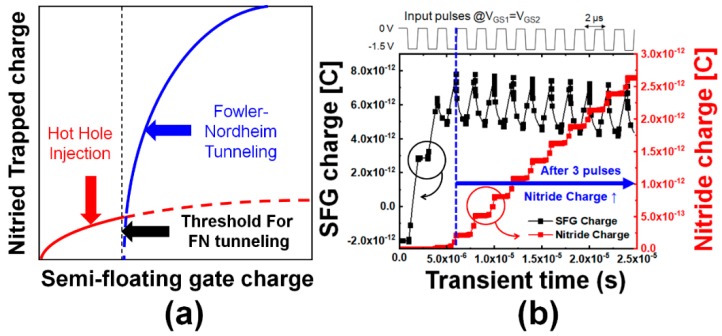
Total charges in the semi-floating gate (SFG) and the nitride regions. (**a**) Qualitative explanation of required holes accumulated in the SFG region to meet the condition of increased probability of injection into the nitride charge-trap layer by Fowler–Nordheim (FN) tunneling; (**b**) Total charges in the SFG (left) and nitride (right) regions after series of potentiating pulses as a function of time obtained by technology computer-aided design (TCAD) simulation.

**Figure 5 micromachines-10-00032-f005:**
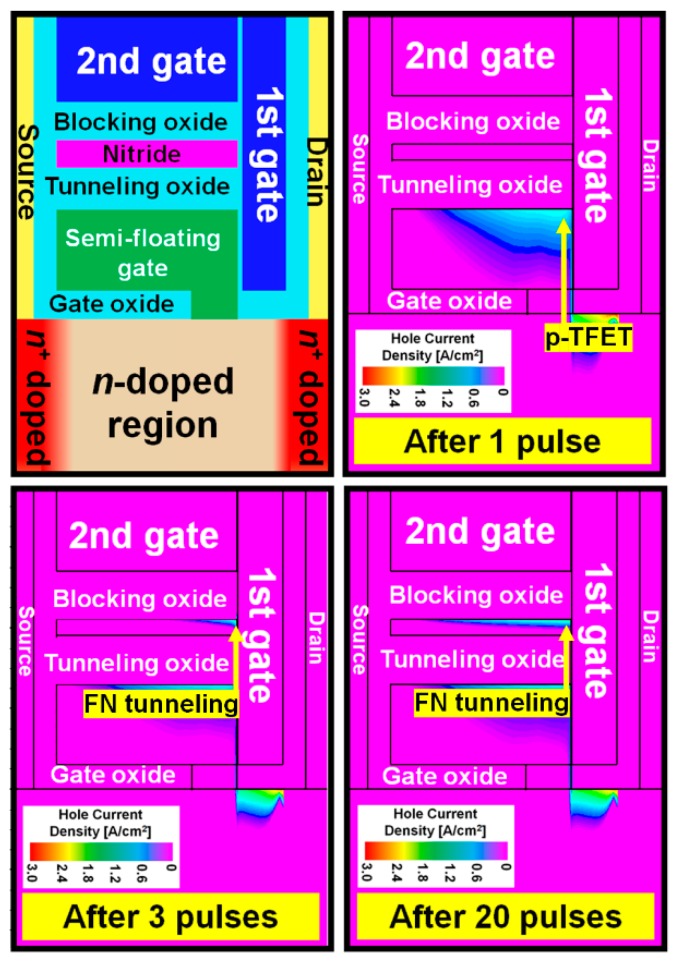
Hole current density after successive potentiation pulses, *V*_GS1_ = *V*_GS2_ = −1.5 V. As the number of pulses increases, the holes at the Fermi distribution tail accumulated in the SFG region see the triangular energy barrier and become more probable for injection into the nitride by FN tunneling.

**Figure 6 micromachines-10-00032-f006:**
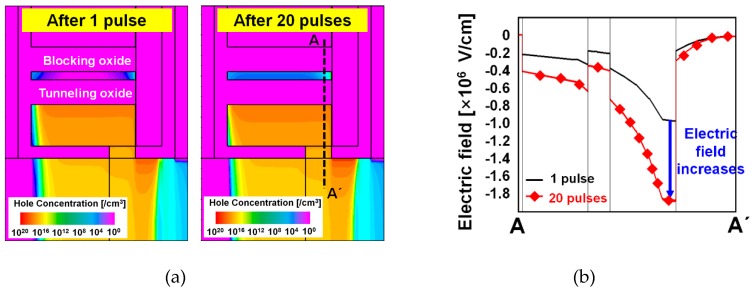
Pulse-number-dependent carrier distribution and electric field. (**a**) Distribution of hole concentration in the SFG and nitride charge-trap layer and (**b**) electric field along the cutline A-A′ in (**a**) after 1 and 20 potentiation pulses.

**Figure 7 micromachines-10-00032-f007:**
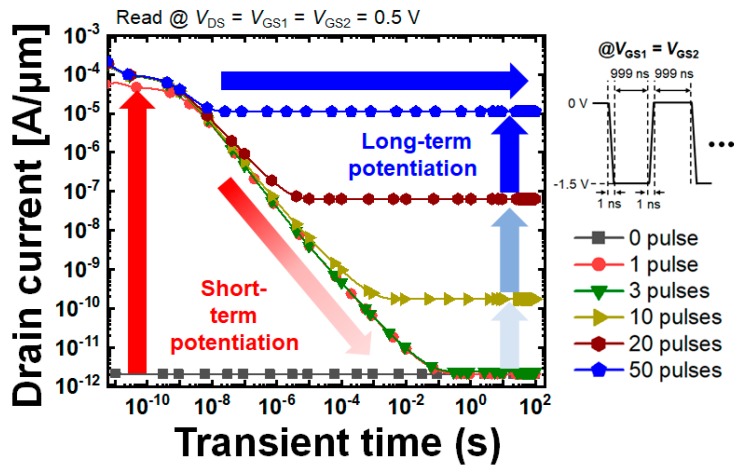
Retention characteristics under a constant read bias condition, *V*_GS1_ = *V*_GS2_ = *V*_DS_ = 0.5 V after potentiating operations with different numbers of potentiation pulses.

**Figure 8 micromachines-10-00032-f008:**
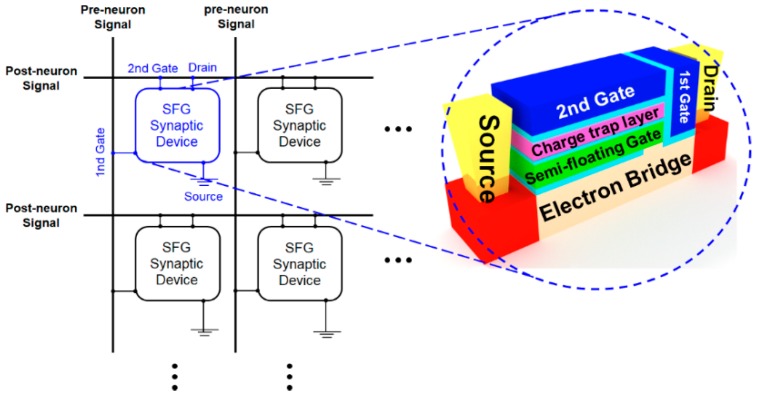
Array architecture for the artificial spike neural network (SNN) based on the proposed SFGST device with the full accommodation of the developed bias schemes.

**Figure 9 micromachines-10-00032-f009:**
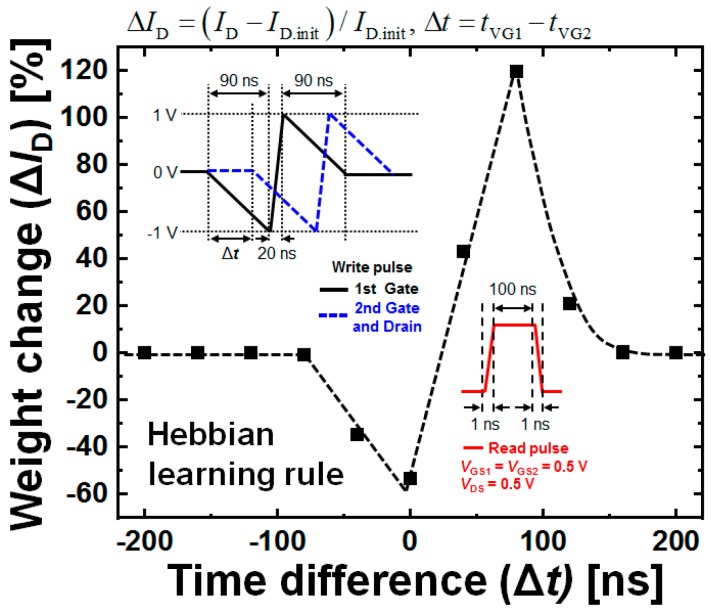
Simulated spike-timing-dependent plasticity (STDP) characteristics of the SFGST after a single triangular spike. Following the Hebbian learning rule [16], the synaptic change is determined by Δ*t*.

**Figure 10 micromachines-10-00032-f010:**
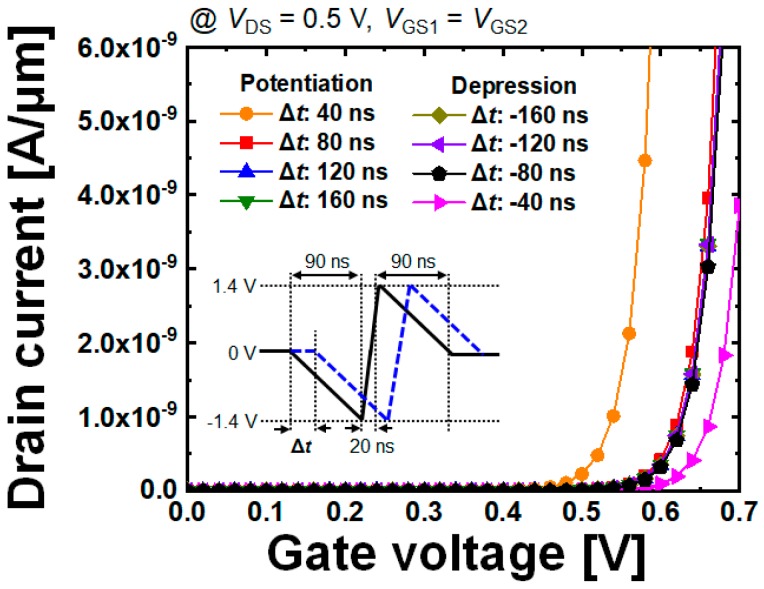
Variation in threshold voltage by the potentiated or depressed SFG. After two pulses are fed with time difference (pre- and post-input signals), potentiation and depression take place under the conditions of Δ*t* > 0 and Δ*t* < 0, respectively. The shorter time interval between the pre-and post-input signals, the larger becomes the variation in the threshold voltage.

**Figure 11 micromachines-10-00032-f011:**
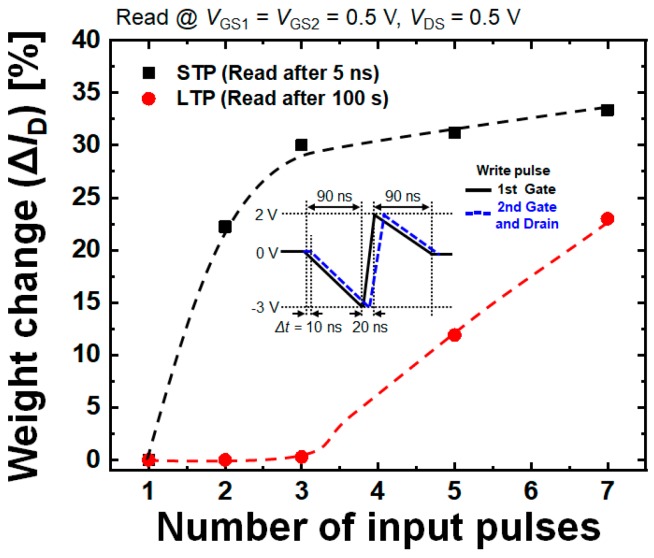
Simulated learning operations of the SFGST according to the number of potentiation pulses.

**Table 1 micromachines-10-00032-t001:** Critical dimensions and process parameters in the designed semi-floating-gate synaptic transistor (SFGST).

Region	Length (nm)	Thickness (nm)	Doping Concen Tration (cm^−3^)
1st Gate	100	37	p-type 1 × 10^20^
2nd Gate	400	10	p-type 1 × 10^20^
SFG	400	10	p-type 1 × 10^18^
Source junction	100	20	n-type 1 × 10^20^
Channel	500	20	n-type 1 × 10^17^
Drain junction	100	20	n-type 1 × 10^20^
Gate oxide	-	3	-
Tunneling oxide	-	6	-
Nitride	-	2	-
Blocking oxide	-	6	-

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
