# Peer review of "Design and Characterization of Semi-Floating-Gate Synaptic Transistor"

_micromachines, 2019, doi:10.3390/mi10010032_

Round 1

Reviewer 1 Report

Dear Authors,

Your work is timely and the characteristics of your device in terms of similarity to synapse behavior is promising.

However, I cannot perform I technical evaluation of your device since you have chosen to plot 

'hole current density'.

The adequate quantity to plot for a charge storage device would be 'hole charge density' as well as 'electron charge density' in a few selected cases. Your device is not fully unipolar hence both charges need to be considered.

Please revise your manuscript both in terms for the figures and also conclusions based on these new plots.

Author Response

Dear the Reviewer 1, 

We the authors deeply thank the reviewer for the times and effort generously shared for reviewing our manuscript. The comments and questions have been carefully read and the answers have been prepared here. Due to the amount of contents and corrected figure, an electronic file for the answer sheet is submitted and hope you to find it below. Thank you so much again. 

- Sincerely, Seongjae Cho, the corresponding author. 

Reviewer 2 Report

Device physics needs to be more clearly established. TCAD could have help but unfortunately hypothesis/models used in TCAD are not detailed which leads to some lack of confidence in the presented results.

Write/Erase mechanisms are not sufficiently established

In figure 4b, Charge extraction method would be welcome. Additional Figure with variation of current vs time would be welcome.

Section 3.1 clearly needs detailed Hypothesis/models used in TCAD; The conclusions drawn in term of the role of ‘trapped charges in the Nitride’ and ‘multi-level states can be realized’ are not really established.

Additional reference to Herb’rule would be welcome.

Could be a good paper but clearly needs large improvement to be published.

Author Response

Dear the Reviewer 2, 

We the authors deeply thank the reviewer for the times and effort generously shared for reviewing our manuscript. The comments and questions have been carefully read and the answers have been prepared here. Due to the amount of contents, an electronic file for the answer sheet is submitted and hope you to find it below. Thank you so much again. 

- Sincerely, Seongjae Cho, the corresponding author. 

Round 2

Reviewer 1 Report

I would you to rephrase the sentencebelow, since I believe that the electric field across the tunneling gate is change by increasing amount of hole charge.

Your simulation does model the fermi tail of the hole distribution since it is using drift-diffusion where carrier energies/temperatures are not included in the equations. All your carriers are at equilibrium with the lattice energy by definition.

Fowler-Nordheim is a field-dependent tunnelling mechanism so that agrees with your modeled results.

In order to clarify this

- Please plot the field across the tunneling oxide for various number of pulses, corresponding to the  cases in Fig. 5

- I would also like to see the hole distribution for the cases in Fig. 5. (even if you choose to not included these plots in the final manuscript, that is optional)

"When adequate amount of holes is charged in the SFG region, the holes in the Fermi distribution tail 97 see the triangular energy barrier, and the probability of injection into the nitride charge-trap layer by 98 Fowler-Nordheim (FN) tunneling is increased as demonstrated in Fig. 4(a)."

Author Response

Dear the Reviewer, 

Thank you for the valuable comments and requests. 

We have completed the 2nd revision and submit the answer sheet. 

Since the answer sheet includes the figures, an independent electronic file is submitted instead of simply filling out the empty box for answers. 

Deeply thank you again for generously sharing your precious time and effort to review our manuscript. 

- Sincerely, Prof. Seongjae Cho, the corresponding author. 
